# VARIATIONAL PSEUDO LABELS FOR META TEST-TIME ADAPTATION

## ABSTRACT

Test-time model adaptation has shown great effectiveness in generalizing over domain shifts. A most successful tactic for test-time adaptation conducts further optimization on the target data using the predictions by the source-trained model. However, due to domain shifts, the source-trained model predictions themselves can be largely inaccurate, which results in a model misspecified to the target data and therefore damages their adaptation ability. In this paper, we address test-time adaptation from a probabilistic perspective. We formulate model adaption as a probabilistic inference problem, which incorporates the uncertainty into source model predictions by modeling pseudo labels as distributions. Based on the probabilistic formalism, we propose variational pseudo labels that explore the information of neighboring target samples to improve pseudo labels and achieve a model better specified to target data. By a meta-learning paradigm, we train our model by simulating domain shifts and the test-time adaptation procedure. In doing so, our model learns the ability to generate more accurate pseudo-label distributions and to adapt to new domains. Experiments on five widely used datasets demonstrate the effectiveness of our proposal.

## 1 INTRODUCTION

Deep neural networks start to exhibit generalizability problems and suffer from performance degradation as soon as test data distributions differ from the ones experienced during training, (Geirhos et al., 2018; Recht et al., 2019). To deal with the distribution shift, domain adaptation, e.g., (Saenko et al., 2010; Long et al., 2015; Lu et al., 2020; Li et al., 2021) and domain generalization, e.g., (Muandet et al., 2013; Motiian et al., 2017; Li et al., 2017; 2020) have proven effective tactics. However, these two settings either require a large number of (unlabeled) target data during training or do not consider any target information during generalization at all. Both of which are not necessarily valid assumptions in realistic scenarios.

Test-time adaptation, e.g., (Sun et al., 2020; Varsavsky et al., 2020; Wang et al., 2021) goes beyond these two setting and introduces a new learning paradigm, which trains a model on source data and further optimizes it using the unlabeled target data at test time to adapt to the target domain. One widely applied strategy for test-time adaptation updates model parameters by self-supervision (Liang et al., 2020; Wang et al., 2021; Iwasawa & Matsuo, 2021; Niu et al., 2022). However, due to domain shifts, the source-model predictions on the target samples can be uncertain and inaccurate. As self-supervision-based test-time adaptation is often achieved by optimization with pseudo labels or entropy minimization based on the source-trained model predictions, the model can be overconfident on some mispredictions. As a result, the adapted model becomes unreliable and misspecified (Wilson & Izmailov, 2020) to the target data.

In this paper we make three contributions. First, we address test-time adaptation in a probabilistic framework by formulating it as a variational inference problem. We define pseudo labels as stochastic variables and estimate a distribution over them by variational inference. By doing so, the uncertainty in source-trained model predictions is incorporated into the adaptation to the target data at test time. Second, thanks to the proposed probabilistic formalism, it is natural and convenient to utilize variational distributions to leverage extra information. By hinging on this benefit, we design the variational pseudo labels to explore the neighboring information of target samples into the inference of the pseudo label distributions. By doing so, the variational pseudo labels are more

accurate, which enables the source-trained model to be better specified to target data and therefore conducive to model adaptation. Third, we adopt a meta-learning paradigm for optimization to simulate test-time adaptation on source domains. More specifically, the model is exposed to domain shifts iteratively and optimized to learn the ability of adapting to unseen domains. We conduct experiments on three widely-used benchmarks to demonstrate the promise and effectiveness of our method for test-time adaptation.

## 2 METHODOLOGY

### 2.1 PRELIMINARY

We were given data from different domains defined on the joint space $\mathcal{X} \times \mathcal{Y}$, where $\mathcal{X}$ and $\mathcal{Y}$ denote the data space and label space, respectively. The domains are split into several source domains $\mathcal{D}_s = \left\{ (\mathbf{x}_s, \mathbf{y}_s)^i \right\}_{i=1}^{N_s}$ and target domains $\mathcal{D}_t = \left\{ (\mathbf{x}_t, \mathbf{y}_t)^i \right\}_{i=1}^{N_t}$. The goal is to train a model on source domains that is expected to generalize well on the (unseen) target domains.

To this end, test-time adaptation methods, e.g., (Wang et al., 2021; Zhang et al., 2021; Niu et al., 2022), have recently been proposed. These methods adapt the source-trained model by optimization to target domains at test time.

A common strategy in these methods is that the model $\boldsymbol{\theta}$ is first trained on source data $\mathcal{D}_s$ by minimizing a supervised loss $\mathcal{L}_{train}(\boldsymbol{\theta}) = \mathbb{E}_{(\mathbf{x}_s, \mathbf{y}_s)^i \in \mathcal{D}_s}[L_{\mathrm{CE}}(\mathbf{x}_s, \mathbf{y}_s; \boldsymbol{\theta})]$; and then at test time they adapt the model $\boldsymbol{\theta}_s$ to the target domain by optimization with certain surrogate losses, e.g., entropy minimization, based on unlabeled test data, which is formulated as:

$$\mathcal{L}_{test}(\boldsymbol{\theta}) = \mathbb{E}_{\mathbf{x}_t \in \mathcal{D}_t}[L_E(\mathbf{x}_t; \boldsymbol{\theta}_s)], \tag{1}$$

where the entropy is calculated on the source model predictions. However, test samples from the target domain could be largely misclassified by the source model due to the domain shift, resulting in large uncertainty in the predictions. Moreover, the entropy minimization tends to update the model with high confidence even for the wrong predictions, which would cause a misspecified model for the target domain.

To solve those problems, in this work we address test-time model adaptation from a probabilistic perspective. We propose a probabilistic inference framework that models the uncertainty of the source-model predictions by defining distributions over pseudo labels. Moreover, under the probabilistic formalism, we propose designing variational pseudo labels, which enables the model to incorporate the neighboring information in test samples to combat domain shifts. We adopt a meta-learning paradigm for optimization, which simulates the domain shifts and adaptation procedure. By doing so, the model learns to acquire the ability to further adapt itself with pseudo labels to unseen target domains. We provide a graphical illustration to highlight the differences between common test-time adaptation and our proposals in Figure 1.

### 2.2 PROBABILISTIC TEST-TIME ADAPTATION WITH LATENT PSEUDO LABELS

We first provide a probabilistic formulation for test-time adaptation based on pseudo labels. Given the target sample $\mathbf{x}_t$ and the source-trained model $\boldsymbol{\theta}_s$, we would like to make predictions on the target sample. To this end, we formulate the predictive likelihood as follows:

$$p(\mathbf{y}_t | \mathbf{x}_t, \boldsymbol{\theta}_s) = \int p(\mathbf{y}_t | \mathbf{x}_t, \boldsymbol{\theta}_t) p(\boldsymbol{\theta}_t | \mathbf{x}_t, \boldsymbol{\theta}_s) d\boldsymbol{\theta}_t \approx p(\mathbf{y}_t | \mathbf{x}_t, \boldsymbol{\theta}_t^*), \tag{2}$$

where we use the value $\boldsymbol{\theta}_t^*$ obtained by the maximum a posterior (MAP) to approximate the integration (Finn et al., 2018). Intuitively, the MAP approximation is interpreted as inferring the posterior over $\boldsymbol{\theta}_t$: $p(\boldsymbol{\theta}_t | \mathbf{x}_t, \boldsymbol{\theta}_s) \approx \delta(\boldsymbol{\theta}_t = \boldsymbol{\theta}_t^*)$, which we obtain by adapting $\boldsymbol{\theta}_s$ using the target data $\mathbf{x}_t$.

To model the uncertainty of predictions for more robust test-time adaptation, we treat pseudo labels as stochastic variables in the probabilistic framework as shown in Figure 1 (b). The pseudo labels are obtained from the source model predictions, which follows categorical distributions. Then we

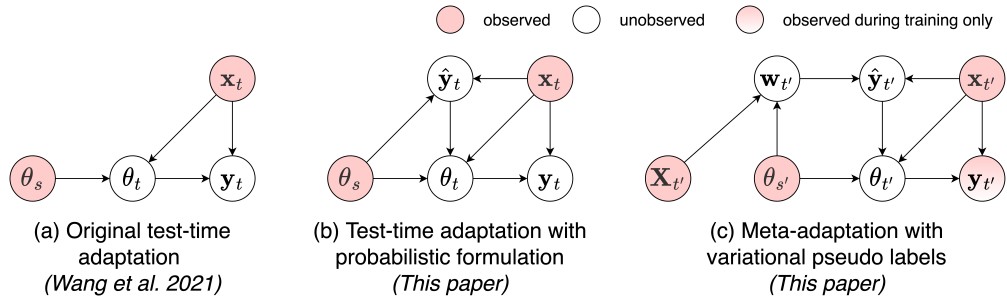

(a) Original test-time adaptation
(Wang et al. 2021)

(b) Test-time adaptation with probabilistic formulation
(This paper)

(c) Meta-adaptation with variational pseudo labels
(This paper)

Figure 1: **Graphical illustrations for test-time adaptation.** (a) The original test-time adaptation algorithm (Wang et al., 2021) obtains an adapted $\boldsymbol{\theta}_t$ by entropy minimization of the unlabeled target data $\mathbf{x}_t$ on source trained model $\boldsymbol{\theta}_s$. (b) Our probabilistic formulation models the uncertainty of pseudo labels $p(\hat{\mathbf{y}}_t)$ for more robust adaptation. (c) Furthermore, we propose meta adaptation with variational pseudo labels to incorporate neighboring target information into pseudo label generation and train the model under the meta-learning setting. Note that $\mathbf{y}_{t'}$ are labels of meta-target data and observed only in training. The actual labels $\mathbf{y}_t$ of target data are unavailable for test-time adaptation at inference time.

reformulate eq. (2) as follows:

$$p(\mathbf{y}_t|\mathbf{x}_t, \boldsymbol{\theta}_s) = \int p(\mathbf{y}_t|\mathbf{x}_t, \boldsymbol{\theta}_t) \Big[ \int p(\boldsymbol{\theta}_t|\hat{\mathbf{y}}_t, \mathbf{x}_t, \boldsymbol{\theta}_s) p(\hat{\mathbf{y}}_t|\mathbf{x}_t, \boldsymbol{\theta}_s) d\hat{\mathbf{y}}_t \Big] d\boldsymbol{\theta}_t$$

$$\tag{3}$$

$$\approx \mathbb{E}_{p(\hat{\mathbf{y}}_t|\mathbf{x}_t, \boldsymbol{\theta}_s)}[p(\mathbf{y}_t|\mathbf{x}_t, \boldsymbol{\theta}_t^*)],$$

where $\boldsymbol{\theta}_t^*$ is the MAP value of $p(\boldsymbol{\theta}_t|\hat{\mathbf{y}}_t, \mathbf{x}_t, \boldsymbol{\theta}_s)$, which is obtained via gradient descent on the data $\mathbf{x}_t$ and the corresponding pseudo labels $\hat{\mathbf{y}}_t$ starting from $\boldsymbol{\theta}_s$. The formulation allows us to sample different pseudo labels from the categorical distribution $p(\hat{\mathbf{y}}_t)$ to adapt the model $\boldsymbol{\theta}_t^*$, which takes into account the uncertainty of predictions by the source-trained model.

By approximating the expectation of $p(\hat{\mathbf{y}}_t)$ with the `argmax` function on $p(\hat{\mathbf{y}}_t)$, $\boldsymbol{\theta}_t^*$ is obtained by gradient descent based on only a point estimation of the pseudo label $p(\hat{\mathbf{y}}_t)$. However, due to domain shifts, the `argmax` value of $p(\hat{\mathbf{y}}_t)$ is not guaranteed to be always correct. The adaptation then is similar to entropy minimization (eq. 1), where the adapted model can achieve high confidence but wrong predictions of some target samples due to domain shifts. For example, consider a toy binary classification task, where the predicted probability is $[0.4, 0.6]$ with the ground-truth label $[1, 0]$. The pseudo label generated by selecting the maximum probability is $[0, 1]$, which is inaccurate. Optimization based on these labels would give rise to a model misspecified to target data, failing to adapt to the target domain.

In contrast, our probabilistic formulation allows us to sample pseudo labels from the categorical distribution $p(\hat{\mathbf{y}}_t|\mathbf{x}_t, \boldsymbol{\theta}_s)$, which incorporates the uncertainty of the pseudo label $\hat{\mathbf{y}}_t$ in a principled way. Again using the above example, the pseudo label sampled from the predicted distribution has a probability of $40\%$ to be the ground-truth label, which leads to the adaptation of the model in the correct direction. Therefore, our formulation achieves better adaptation by accessing accurate pseudo labels from the inaccurate prediction distributions.

## 2.3 VARIATIONAL PSEUDO LABELS

Under the probabilistic formalism, we derive variational inference of pseudo labels. To train the ability of the model to generate better variational pseudo labels and to fully utilize the pseudo label distributions for better adaptation, we adopt the meta-learning paradigm to simulate domain shifts and test-time adaptation procedures (Finn et al., 2017; Dou et al., 2019; Xiao et al., 2022). We split the source domains $\mathcal{D}_s$ into meta-source domains $\mathcal{D}_{s'}$ and a meta-target domain $\mathcal{D}_{t'}$ during training. The meta-target domain is selected randomly in each iteration to mimic diverse domain shifts.

To simulate the test-time adaptation and estimation procedure, we maximize the log-likelihood of the meta-target samples after model adaptation on the meta-target data:

$$\log p(\mathbf{y}_{t'}|\mathbf{x}_{t'}, \boldsymbol{\theta}_{s'}) = \log \int p(\mathbf{y}_{t'}|\mathbf{x}_{t'}, \boldsymbol{\theta}_{t'}) \Big[ \int p(\boldsymbol{\theta}_{t'}|\hat{\mathbf{y}}_{t'}, \mathbf{x}_{t'}, \boldsymbol{\theta}_{s'}) p(\hat{\mathbf{y}}_{t'}|\mathbf{x}_{t'}, \boldsymbol{\theta}_{s'}) d\hat{\mathbf{y}}_{t'} \Big] d\boldsymbol{\theta}_{t'}$$

$$\approx \log \int p(\mathbf{y}_{t'}|\mathbf{x}_{t'}, \boldsymbol{\theta}_{t'}^*) p(\hat{\mathbf{y}}_{t'}|\mathbf{x}_{t'}, \boldsymbol{\theta}_{s'}) d\hat{\mathbf{y}}_{t'} \tag{4}$$

$$\geq \mathbb{E}_{p(\hat{\mathbf{y}}_{t'}|\mathbf{x}_{t'}, \boldsymbol{\theta}_{s'})}[\log p(\mathbf{y}_{t'}|\mathbf{x}_{t'}, \boldsymbol{\theta}_{t'}^*)],$$

where $p(\hat{\mathbf{y}}_{t'}|\mathbf{x}_{t'}, \boldsymbol{\theta}_{s'})$ denotes the distribution of pseudo labels generated by the meta-source model $\boldsymbol{\theta}_{s'}$ on the meta-target data $\mathbf{x}_{t'}$. $\boldsymbol{\theta}_{t'}^*$ is the MAP value of $p(\boldsymbol{\theta}_{t'}|\hat{\mathbf{y}}_{t'}, \mathbf{x}_{t'}, \boldsymbol{\theta}_{s'})$ similar to eq. (3), which is learned to mimic the test-time adaptation procedure.

Under the meta-learning setting, the actual labels $\mathbf{y}_{t'}$ of the meta-target data is accessible since source data are fully labeled as shown in Figure 1 (c). We then simulate the test evaluation procedure and further supervise the adapted model $\boldsymbol{\theta}_{t'}^*$ on its meta-target predictions by the actual labels.

The maximization of the log-likelihood of $p(\mathbf{y}_{t'}|\mathbf{x}_{t'}, \boldsymbol{\theta}_{t'}^*)$ is realised by a cross-entropy loss on the meta-target predictions and the actual meta-target labels. Intuitively, the pseudo-label adapted model is supervised to achieve good performance on the adapted data. Thus, it learns the ability to generate better pseudo labels and achieve better adaptation across domain shifts with these pseudo labels on new unseen domains.

**Variational pseudo labels.** Moreover, we also propose variational pseudo labels that incorporate information of the neighboring target samples to estimate pseudo label distributions that are more robust against domain shifts. The variational pseudo labels is natural and convenient to deployed under the probabilistic formulation. Assume that we have a batch of meta-target data $\mathbf{X}_{t'} = \{\mathbf{x}_{t'}^i\}_{i=1}^M$, we reformulate eq. (4) as:

$$\log p(\mathbf{y}_{t'}|\mathbf{x}_{t'}, \boldsymbol{\theta}_{s'}, \mathbf{X}_{t'})$$

$$= \log \int p(\mathbf{y}_{t'}|\mathbf{x}_{t'}, \boldsymbol{\theta}_{t'}) \Big[ \int \int p(\boldsymbol{\theta}_{t'}|\hat{\mathbf{y}}_{t'}, \mathbf{x}_{t'}, \boldsymbol{\theta}_{s'}) p(\hat{\mathbf{y}}_{t'}, \mathbf{w}_{t'}|\mathbf{x}_{t'}, \boldsymbol{\theta}_{s'}, \mathbf{X}_{t'}) d\hat{\mathbf{y}}_{t'} d\mathbf{w}_{t'} \Big] d\boldsymbol{\theta}_{t'} \tag{5}$$

$$= \log \int \int p(\mathbf{y}_{t'}|\mathbf{x}_{t'}, \boldsymbol{\theta}_{t'}^*) p(\hat{\mathbf{y}}_{t'}, \mathbf{w}_{t'}|\mathbf{x}_{t'}, \boldsymbol{\theta}_{s'}, \mathbf{X}_{t'}) d\hat{\mathbf{y}}_{t'} d\mathbf{w}_{t'} d\boldsymbol{\theta}_{t'},$$

where $\boldsymbol{\theta}_{t'}^*$ is the MAP value of $p(\boldsymbol{\theta}_{t'}|\hat{\mathbf{y}}_{t'}, \mathbf{x}_{t'}, \boldsymbol{\theta}_{s'})$. We introduce the latent variable $\mathbf{w}_{t'}$ to integrate the information of the neighboring target samples $\mathbf{X}_{t'}$ as shown in Figure 1.

To approximate the true posterior of the joint distribution $p(\hat{\mathbf{y}}_{t'}, \mathbf{w}_{t'})$, we introduce a variational posterior $q(\hat{\mathbf{y}}_{t'}, \mathbf{w}_{t'}|\mathbf{x}_{t'}, \boldsymbol{\theta}_{s'}, \mathbf{X}_{t'}, \mathbf{Y}_{t'})$, where $\mathbf{Y}_{t'} = \{\mathbf{y}_{t'}^i\}_{i=1}^M$ denotes the actual labels of the meta-target data $\mathbf{X}_{t'}$. To facilitate the estimation of pseudo labels, we set the prior distribution as:

$$p(\hat{\mathbf{y}}_{t'}, \mathbf{w}_{t'}|\mathbf{x}_{t'}, \boldsymbol{\theta}_{s'}, \mathbf{X}_{t'}) = p(\hat{\mathbf{y}}_{t'}|\mathbf{w}_{t'}, \mathbf{x}_{t'}) p_\phi(\mathbf{w}_{t'}|\boldsymbol{\theta}_{s'}, \mathbf{X}_{t'}) \tag{6}$$

where $p_\phi(\mathbf{w}_{t'}|\boldsymbol{\theta}_{s'}, \mathbf{X}_{t'})$ is generated by the features of $\mathbf{X}_{t'}$ together with their output values based on $\boldsymbol{\theta}_{s'}$.

Similarly, we define the variational posterior distribution as:

$$q(\hat{\mathbf{y}}_{t'}, \mathbf{w}_{t'}|\mathbf{x}_{t'}, \boldsymbol{\theta}_{s'}, \mathbf{X}_{t'}, \mathbf{Y}_{t'}) = p(\hat{\mathbf{y}}_{t'}|\mathbf{w}_{t'}, \mathbf{x}_{t'}) q_\phi(\mathbf{w}_{t'}|\boldsymbol{\theta}_{s'}, \mathbf{X}_{t'}, \mathbf{Y}_{t'}). \tag{7}$$

where $q_\phi(\mathbf{w}_{t'}|\boldsymbol{\theta}_{s'}, \mathbf{X}_{t'}, \mathbf{Y}_{t'})$ is obtained by the features of $\mathbf{X}_{t'}$ and the actual labels $\mathbf{Y}_{t'}$ based on $\boldsymbol{\theta}_{s'}$.

By introducing eqs. (6) and (7) into (5), we derive the evidence lower bound (ELBO) of the log-likelihood in eq. (5) as follows:

$$\log p(\mathbf{y}_{t'}|\mathbf{x}_{t'}, \boldsymbol{\theta}_{s'}, \mathbf{X}_{t'}) \geq \mathbb{E}_{q_\phi(\mathbf{w}_{t'})} \mathbb{E}_{p(\hat{\mathbf{y}}_{t'}|\mathbf{w}_{t'}, \mathbf{x}_{t'})}[\log p(\mathbf{y}_{t'}|\mathbf{x}_{t'}, \boldsymbol{\theta}_{t'}^*)]$$

$$- \mathbb{D}_{KL}[q_\phi(\mathbf{w}_{t'}|\boldsymbol{\theta}_{s'}, \mathbf{X}_{t'}, \mathbf{Y}_{t'}) || p_\phi(\mathbf{w}_{t'}|\boldsymbol{\theta}_{s'}, \mathbf{X}_{t'})]. \tag{8}$$

Rather than directly using the meta-source model $\boldsymbol{\theta}_{s'}$, we estimate the pseudo labels $\mathbf{y}_{t'}$ from the latent variable $\mathbf{w}_{t'}$, which integrates the features of neighboring target samples. By considering the actual labels $\mathbf{Y}_{t'}$, the variational distribution utilizes both the target information and categorical information of the neighboring samples. Thus, the variational posterior models the distribution of different categories in the target domain more reliably and produces more accurate pseudo labels to improve model adaptation.

## 2.4 META TEST-TIME ADAPTATION: TRAINING AND INFERENCE

To mimic domain shifts during training, we split each iteration into meta-source, meta-adaptation, and meta-target to simulate the training stage on source domains, test-time adaptation, and test stage on target data, respectively.

Under the meta-learning paradigm, the model is iteratively exposed to domain shifts and learns the capability to adapt the meta-source model $\boldsymbol{\theta}_{s'}$ to meta-target MAP $\boldsymbol{\theta}_{t'}^*$ with the variational pseudo labels. The parameters in the variational inference model $\phi$ are jointly optimized in order to generate better pseudo labels against domain shifts.

**Meta-source.** We first train and optimize the model on meta-source domains by minimizing the supervised loss:

$$\boldsymbol{\theta}_{s'} = \min_{\boldsymbol{\theta}} \mathbb{E}_{(\mathbf{x}_{s'}, \mathbf{y}_{s'})) \in \mathcal{D}_{s'}} [L_{\text{CE}}(\mathbf{x}_{s'}, \mathbf{y}_{s'}; \boldsymbol{\theta})], \tag{9}$$

where $(\mathbf{x}_{s'}, \mathbf{y}_{s'})$ denotes the input-label pairs of samples on meta-source domains. $\boldsymbol{\theta}_{s'}$ are the model parameters trained on the meta-source data.

**Meta-adaptation.** Once the meta-source-trained model $\boldsymbol{\theta}_{s'}$ is obtained, we generate the pseudo labels $p(\hat{\mathbf{y}}_{t'}|\mathbf{w}_{t'}, \mathbf{x}_{t'})$ of the meta-target data with the variational posterior $q_{\phi}(\mathbf{w}_{t'}|\boldsymbol{\theta}_{s'}, \mathbf{X}_{t'}, \mathbf{Y}_{t'})$. The test-time adaptation procedure is simulated by obtaining $\boldsymbol{\theta}_{t'}^*$:

$$\boldsymbol{\theta}_{t'}^* = \boldsymbol{\theta}_{s'} - \lambda_1 \nabla_{\boldsymbol{\theta}} L_{\text{CE}}(\mathbf{x}_{t'}, \hat{\mathbf{y}}_{t'}; \boldsymbol{\theta}_{s'}) \qquad \hat{\mathbf{y}}_{t'} \sim p(\hat{\mathbf{y}}_{t'}|\mathbf{w}_{t'}, \mathbf{x}_{t'}), \tag{10}$$

where $\lambda_1$ denotes the learning rate of the optimization in the meta-adaptation stage.

**Meta-target.** Since our final goal is to obtain good performance on the target data after optimization with pseudo labels, we further mimic the test-time inference on the meta-target domain and supervise the meta-target prediction on $\boldsymbol{\theta}_{t'}^*$ by maximizing the log-likelihood, which is equal to minimizing:

$$\mathcal{L}_{meta} = \mathbb{E}_{(\mathbf{x}_{t'}, \mathbf{y}_{t'}) \in \mathcal{D}_{t'}} [\mathbb{E}_{q_{\phi}(\mathbf{w}_{t'})} \mathbb{E}_{p(\hat{\mathbf{y}}_{t'}|\mathbf{w}_{t'}, \mathbf{x}_{t'})} L_{\text{CE}}(\mathbf{x}_{t'}, \mathbf{y}_{t'}; \boldsymbol{\theta}_{t'}^*)] + \mathbb{D}_{KL}[q_{\phi}(\mathbf{w}_{t'}) || p_{\phi}(\mathbf{w}_{t'})], \tag{11}$$

where $\mathbf{y}_{t'}$ denotes the ground truth label of $\mathbf{x}_{t'}$. The parameters $\boldsymbol{\theta}$ are finally updated by

$$\boldsymbol{\theta} = \boldsymbol{\theta}_{s'} - \lambda_2 \nabla_{\boldsymbol{\theta}} \mathcal{L}_{meta}, \tag{12}$$

where $\lambda_2$ denotes the learning rate for the meta-target stage. Note that the loss in eq. (11) is computed on the parameters $\boldsymbol{\theta}_{t'}^*$ obtained by eq. (10), while the optimization is performed over the meta-source-trained parameters $\boldsymbol{\theta}_{s'}$ in eq. (12). Intuitively, the parameters are optimized to learn the ability to handle domain shifts, such that adaptation with variational pseudo labels of data from new domains improves the predictions on the new domain.

The parameters in the variational inference model $\phi$ are jointly trained with $\boldsymbol{\theta}$. To guarantee that the variational pseudo labels do extract the neighboring information for discrimination, we add a cross-entropy loss $\mathcal{L}_{\hat{c}e}$ on the variational pseudo labels and the corresponding actual labels during training. Thus, $\phi$ is updated by:

$$\phi = \phi - \lambda_3 (\nabla_{\phi} \mathcal{L}_{\hat{c}e} - \nabla_{\phi} \mathcal{L}_{meta}), \tag{13}$$

where $\lambda_3$ denotes the learning rates.

**Test-time adaptation and prediction.** At test time, the model trained on the source domains with the above meta-learning strategy $\boldsymbol{\theta}_s$ is adapted by further optimization using eq. (10).

The adapted model is then evaluated on the (unseen) target data $\mathcal{D}_t$. The prediction is formulated as:

$$p(\mathbf{y}_t|\mathbf{x}_t, \boldsymbol{\theta}_s, \mathbf{X}_t) = \int p(\mathbf{y}_t|\mathbf{x}_t, \boldsymbol{\theta}_t) \Big[ \int p(\boldsymbol{\theta}_t|\hat{\mathbf{y}}_t, \mathbf{x}_t, \boldsymbol{\theta}_s) p(\hat{\mathbf{y}}_t, \mathbf{w}_t|\mathbf{x}_t, \boldsymbol{\theta}_s, \mathbf{X}_t) d\hat{\mathbf{y}}_t d\mathbf{w}_t \Big] d\boldsymbol{\theta}_t$$
$$= \mathbb{E}_{p_{\phi}(\mathbf{w}_t)} \mathbb{E}_{p(\hat{\mathbf{y}}_t|\mathbf{w}_t, \mathbf{x}_t)} [\log p(\mathbf{y}_t|\mathbf{x}_t, \boldsymbol{\theta}_t^*)], \tag{14}$$

where $\boldsymbol{\theta}_t^*$ is the MAP value of $p(\boldsymbol{\theta}_t|\mathbf{x}_t, \hat{\mathbf{y}}_t, \boldsymbol{\theta}_s)$. $p_{\phi}(\mathbf{w}_t) = p_{\phi}(\mathbf{w}_t|\boldsymbol{\theta}_s, \mathbf{X}_t)$ is generated by the features of $\mathbf{X}_t$ according to the outputs or common pseudo labels based on $\boldsymbol{\theta}_s$.

## 3 RELATED WORK

**Test-time adaptation.** By combining the advantages of both domain adaptation (Ganin & Lempitsky, 2015; Long et al., 2015; Hoffman et al., 2018; Lu et al., 2020; Tzeng et al., 2017; Shen et al.,

2022) and domain generalization (Muandet et al., 2013; Li et al., 2017; 2019; Du et al., 2020; Zhou et al., 2020; 2022), test-time adaptation (Sun et al., 2020; Dubey et al., 2021; Wang et al., 2021; Zhou & Levine, 2021; Chen et al., 2022) and source-free adaptation (Liang et al., 2020; Eastwood et al., 2021) are proposed to train a model only on source domains while adapting it to unlabeled target data at test time. Several methods update normalization statistics of the model to handle domain shifts (Schneider et al., 2020; Du et al., 2021; Hu et al., 2021). Sun et al. (2020) proposed to fine-tune the model parameters by a self-supervised loss at test time. Liu et al. (2021) further enhanced the method by introducing a test-time feature alignment strategy. Instead of using an extra self-supervised loss, Wang et al. (2021) proposed fully test-time adaptation by entropy minimization, which is followed in several recent works (Zhang et al., 2021; Niu et al., 2022; Jang & Chung, 2022). Zhang et al. (2021) minimized the entropy of the marginal output distribution averaged over multiple augmentations of a single target sample. Niu et al. (2022) proposed an efficient test-time adaptation method without forgetting by adapting with low entropy samples with Fisher regularization. Different from these methods, Iwasawa & Matsuo (2021) tried to adjust the classifier with pseudo labels of the target data, without fine-tuning the model parameters. Different from these methods, we propose a probabilistic formulation of test-time adaptation, which models the uncertainty of pseudo labels for better adaptation. We also introduce variational pseudo labels with meta-adaptation to further learn the ability to improve the pseudo labels and adaptations.

**Meta-learning.** Meta-learning-based methods (Alet et al., 2021; Xiao et al., 2022; Goyal et al., 2022) have been studied for test-time adaptation before. Alet et al. (2021) learned to adapt with a contrastive loss. Goyal et al. (2022) meta-learned the loss functions of the test-time adaptation for better adaptation. Xiao et al. (2022) proposed a single sample generalization, which adapts the model to each individual target sample through mimicking domain shifts during training. Our method also learns the adaptation ability under the meta-learning setting. We design meta-adaptation to simulate the test-time adaptation procedure and supervise it based on our probabilistic formulation. We further supervise the meta-adapted model in the meta-target stage to learn the variational pseudo label generation and adaptation ability.

**Pseudo label learning.** In pseudo label learning, the motive is to use a model's best predictions, e.g., high confidence predictions, and use the corresponding samples and their predictions for retraining the model on a given downstream task. Tasks examples include classification (Yalniz et al., 2019; Xie et al., 2020), segmentation (Zou et al., 2020), and object detection (Li et al., 2022). Pham et al. (2021) addresses the problem of confirmation bias in pseudo labeling and utilize a teacher-student network for the task of image classification. (Zou et al., 2019) utilize the output logits of softmax as the prediction probability and train the model to directly maximize the logits. (Wang et al., 2022) utilizes confidence to approximate the domain difference and applies augmentations to improve quality if below the threshold to update the student model. (Rizve et al., 2021) assumes that there is access to target labels in a semi-supervised learning and relies on prediction uncertainties leveraged through the labeled samples in target to and generates new pseudo labels. For semi-supervised learning, (Miyato et al., 2018) proposes a regularization method that uses target data without labels. Shu et al. (2018) constructs a new source domain using pseudo labelling using available target data during source training. For domain generalization and offline adaptation, Abdo et al. (2009) utilize pseudo labels from network extracting features and depth features. For online adaptation, Chen et al. (2022) utilize a contrastive method and utilize pseudo labels to ensure that same class negative samples are not used in contrastive loss optimization. For semi-supervised adaptation Zhou et al. (2021) utilize pseudo labels and use confidence as a criteria. In contrast to existing works, our method uses meta learning and is probabilistic in nature for test-time adaptation.

## 4 EXPERIMENTS

### 4.1 SETTINGS

**Five datasets.** We demonstrate the effectiveness of our method on image classification and domain generalization settings. We evaluate our method on five widely used datasets in domain generalization. *PACS* (Li et al., 2017) consists of 7 classes and 4 domains: Photo, Art painting, Cartoon, and Sketch with 9991 samples. *VLCS* (Fang et al., 2013) consists of 5 classes and 4 domains: Pascal, LabelMe, Caltech, SUN with 10,729 samples. *TerraIncognita* (Beery et al., 2018) consists of 10 classes and 4 domains: Location 100, Location 38, Location 43 and Location 46 with 24,778

Table 1: **Benefits of our probabilistic test-time adaptation.** The experiments are conducted on PACS based on ResNet-18. Our probabilistic formulation achieves better performance than the common test-time adaptation in both online and offline settings. The variational pseudo labels with meta adaptation further improve the overall performance.

| Method | Photo | Art-painting | Cartoon | Sketch | Mean |
|---|---|---|---|---|---|
| *Online Adaptation* | | | | | |
| Test-time adaptation | 93.91 | 78.52 | 78.33 | **74.37** | 81.28 |
| Our probabilistic test-time adaptation | 94.55 | 80.07 | 79.14 | 74.29 | 82.01 |
| *Our method* | **95.50** ±0.2 | **82.90** ±0.4 | **81.28** ±0.5 | 74.11 ±0.7 | **83.45** ±0.2 |
| *Offline Adaptation* | | | | | |
| Test-time adaptation | 94.61 | 80.13 | 79.39 | 74.96 | 82.27 |
| Our probabilistic test-time adaptation | 95.15 | 81.15 | 80.22 | **75.42** | 82.99 |
| *Our method* | **95.80** ±0.5 | **84.32** ±0.8 | **83.44** ±0.4 | 74.57 ±0.2 | **84.78** ±0.36 |

samples. We follow the training and validation split in (Li et al., 2017) and evaluate the model according to the "leave-one-out" protocol (Li et al., 2019; Carlucci et al., 2019). We also evaluate our method on the *Rotated MNIST* and *Fashion-MNIST* datasets following Piratla et al. (2020), where the images are rotated by different angles as different domains. We use the subsets with rotation angles from $15°$ to $75°$ in intervals of $15°$ as five source domains, and images rotated by $0°$ and $90°$ as the target domains.

**Two adaptation settings.** To demonstrate the effectiveness of our method, we evaluate on both *offline* and *online* test-time adaptation settings. *Offline test-time adaptation* assumes there are already unlabeled target data available for adaptation. To be close to real-world application where it is difficult to access the whole target set, we use different amounts of target data for offline adaptation. We adapt the model on the available target data and evaluate it on the entire target set without continuously fine-tuning on the entire target set. We also evaluate our method for *online test-time adaptation* (Iwasawa & Matsuo, 2021). In real-world application, as aforementioned, instead of accessing the entire target set, we usually obtain unlabeled target data in an online manner. To achieve continuous adaptation and improvement of the model on target data, we increment the target data iteratively and keep adapting and evaluating the model on the online target data.

**Implementation details** We make use of ResNet-18 for all our experiments and ablation studies and report the accuracies on ResNet-50 for comparison as well. The backbones are pretrained on ImageNet same as the previous methods. During training, we use a varied learning rate throughout the model. We set the learning rate for the pretrained ResNet to 5e-5 and the learning rate of the variational module and classifiers as 1e-4 for all datasets. During test-time adaptation, we utilize a learning rate of 1e-4 for all layers. We intend to release the code on acceptance of this paper.

## 4.2 ABLATION STUDIES

**Benefits of our probabilistic test-time adaptation** We first investigate the effectiveness of our probabilistic formulation of test-time adaptation and its meta-learned variational pseudo labels. To demonstrate the benefits of the probabilistic formulation, we conduct test-time adaptation with eq. (3) and compare it with a common test-time adaptation tactic (Wang et al., 2021), as in eq. (1). As shown in Table 1, both adaptation with and without probabilistic formulation achieve good improvements over the ERM baseline. Our probabilistic test-time adaptation performs better than the common one on most target domains for both the online and offline adaptation settings, which demonstrates the benefits of modeling uncertainty during adaptation at test time.

Moreover, we incorporate the distribution of pseudo labels into the probabilistic formulation, and further propose the variational pseudo labels with meta-adaptation based on the pseudo label distributions. As shown in the fourth and last row in Table 1, our method further improves the performance of both online and offline adaptation, demonstrating the effectiveness of the variational pseudo labels with meta-adaptation. With the probabilistic formulation, it is natural and simple to define the problem as a variational inference problem and solve the problem under the meta-learning

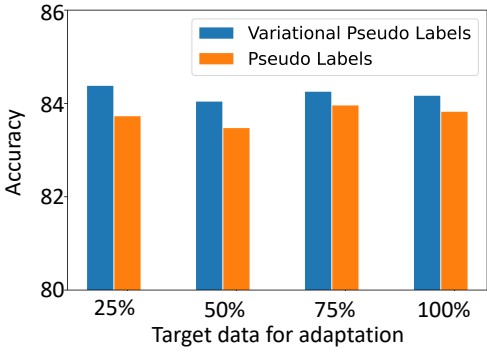 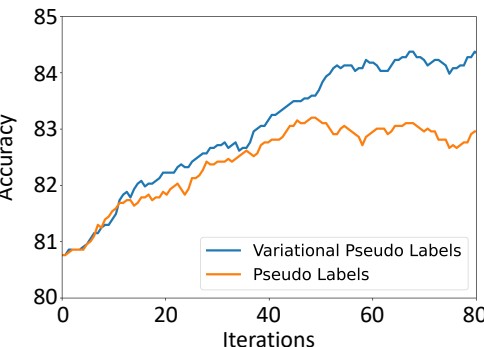

Figure 2: **Benefits of variational pseudo labels.** The experiments are conducted on PACS with ResNet-18 as the backbone. We compare our variational pseudo labels with common pseudo labels with different amounts of adaptation data under the offline settings (left figure). Our variational pseudo labels achieve better overall accuracy consistently. We also provide the accuracy along with adaptation steps on *art-painting* in the right figure. Our method adapts faster and achieves better performance than using the common pseudo labels.

Table 2: Ablation of meta-learning setting: We conduct the below experiments on PACS using ResNet-18. In the first framework, we do not use meta-learning across all stages. We observe that our method performs better on all domains comparatively.

| Settings | Photo | Art-painting | Cartoon | Sketch | *Mean* |
|---|---|---|---|---|---|
| W/o meta-learning | 94.76 ±1.0 | 80.7 ±0.5 | 78.87 ±0.7 | 68.39 ±0.5 | 80.68 |
| *W/ meta-learning* | **95.80** ±0.5 | **84.32** ±0.8 | **83.44** ±0.4 | **74.57** ±0.2 | **84.78** ±0.36 |

framework. The results further demonstrate the benefits and importance of our probabilistic formulations of test-time adaptation.

**Benefits of variational pseudo-labels**  Based on the probabilistic formulation, we introduce variational pseudo labels to incorporate the target information of the neighboring target data for each target sample. To demonstrate the effectiveness of our variational pseudo label, we compare it with the normal pseudo labels drawn directly from the prediction distributions of source-trained models. We evaluate the methods in offline adaptation settings with different amounts of target data. As shown in Figure 2 (left), adaptation with our variational pseudo labels achieves better overall results than the normal pseudo labels consistently. We also provide the adaptation results along with adaptation steps in Figure 2 (right). Starting from the same baseline accuracy, the variational pseudo labels achieve faster adaptation than the normal pseudo labels. Adaptation with variational pseudo labels is less prone to saturating in performance, leading to better final accuracy.

**Benefits of meta learning**  We also investigate the importance of meta-learning in our method. For this experiment, we do not make meta-learning for source training and test-time adaptation and only use variational labels that have been generated for adaptation. We observe that meta-learning indeed helps in our formulation. Without meta-learning, it is difficult for the model to learn the ability to handle domain shifts. Thus, there is a significant decrease in accuracy as shown in Table 2.

**Offline adaptation with less target data**  In real applications, it is difficult to access the entire target set at once for adaptation. Therefore, we estimate our method under the offline test-time adaptation settings with different amounts of target data. As shown in Figure 3, the accuracy increases obviously with small numbers of target samples, e.g., 10% and 25%. However, the overall accuracy tends to saturate when keeping increasing the number of target data for adaptation. This indicates that our method is able to achieve good adaptation under the offline adaptation setting with even small amounts of target data, showing the applicability of the proposed method in practice.

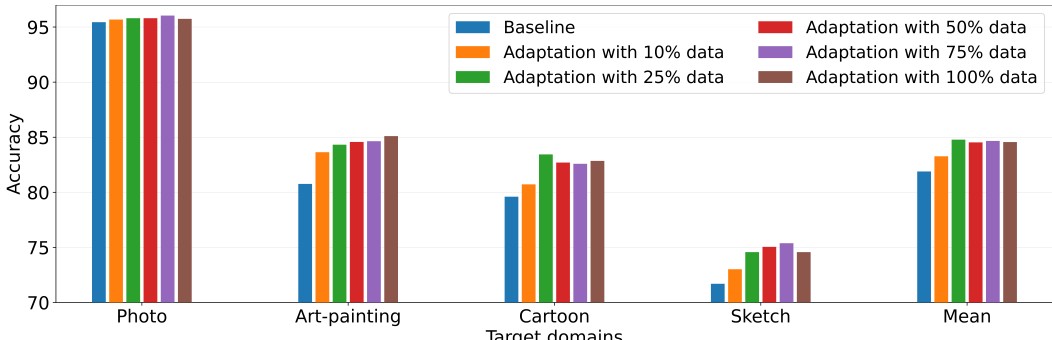

Figure 3: **Offine adaptation with less target data.** The experiments are conducted on PACS using ResNet-18 averaged over five runs. Under the offline-adaptation setting we observe that with increments in the amount of test data samples the accuracy increases steadily for individual domains.

Table 3: **Comparisons on common DG datasets.** The experiments are conducted on all datasets averaged over five runs. We provide the results of our method for the online setting. Our method also performs better than the state-of-the-art domain generalization methods across all datasets.

| Algorithm | PACS | | VLCS | TerraIncognita |
|---|---|---|---|---|
| | ResNet-18 | ResNet-50 | ResNet-18 | ResNet-18 |
| ERM | 79.29 | 83.21 | 74.88 | 40.62 |
| *Standard DG Methods* | | | | |
| MASF (Dou et al., 2019) | 81.0 | 82.7 | - | - |
| ER (Zhao et al., 2020) | 81.5 | 85.3 | 74.4 | - |
| *Test-time Adaptation Methods* | | | | |
| Tent-BN (Wang et al., 2021) | 81.3 | 83.7 | 61.3 | 39.8 |
| SHOT (Liang et al., 2020) | 82.4 | 84.1 | 65.2 | 33.5 |
| T3A (Iwasawa & Matsuo, 2021) | 81.7 | 84.5 | 76.5 | 41.6 |
| TAST (Jang & Chung, 2022) | 81.9 | 84.1 | 77.3 | 42.6 |
| *Our Method* | **83.4** ±0.4 | **85.5** ±0.3 | **77.8** ±0.8 | **46.2** ±0.4 |

## 4.3 COMPARISONS

To further demonstrate the effectiveness of our method, we compare our method with some state-of-the-art test-time adaptation methods and standard domain generalization methods. Table 3 shows the results on PACS, VLCS and TerraIncognita using ResNet-18. Compared with the other state-of-the-art domain generalization methods and test-time adaptation methods with the online adaptation setting, our method performs better. On datasets such as PACS and TerraIncognita, we are better by state-of-the-art methods significantly. We also report results on the PACS dataset using ResNet-50 as the backbone, the performance of our method is competitive and better than most of the state-of-the-art methods. We provide the detailed comparison in Appendix 5.

## 5 CONCLUSION

We propose to cast test-time adaptation as a probabilistic inference problem and model pseudo-labels as distributions in the formulation. By modeling the uncertainty into the pseudo label distributions, the probabilistic formulation mitigates adaptation with inaccurate pseudo labels or predictions, which arises due to domain shifts and lead to misspecified models after adaptation. Based on the probabilistic formulation, we further propose variational pseudo labels under the meta-adaptation paradigm, which exposes the model to domain shifts and learns the ability to adapt with pseudo labels incorporating target information of the neighboring target samples. Ablation studies and further comparisons show the effectiveness of our method on five common domain generalization datasets.

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

## A   DETAILED FORMULATION

We start the objective function from $p(\mathbf{y}_{t'}|\mathbf{x}_{t'}, \boldsymbol{\theta}_{s'}, \mathbf{X}_{t'})$. Here we provide the detailed generating process of the formulation:

$$p(\mathbf{y}_{t'}|\mathbf{x}_{t'}, \boldsymbol{\theta}_{s'}, \mathbf{X}_{t'}) = \int p(\mathbf{y}_{t'}|\mathbf{x}_{t'}, \boldsymbol{\theta}_{t'})p(\boldsymbol{\theta}_{t'}|\mathbf{x}_{t'}, \boldsymbol{\theta}_{s'}, \mathbf{X}_{t'})d\boldsymbol{\theta}_{t'}. \tag{15}$$

We then introduce the pseudo labels $\hat{\mathbf{y}}_{t'}$ as the latent variable into eq. (15) and derive it as:

$$p(\mathbf{y}_{t'}|\mathbf{x}_{t'}, \boldsymbol{\theta}_{s'}, \mathbf{X}_{t'}) = \int p(\mathbf{y}_{t'}|\mathbf{x}_{t'}, \boldsymbol{\theta}_{t'}) \int p(\boldsymbol{\theta}_{t'}|\hat{\mathbf{y}}_{t'}, \mathbf{x}_{t'}, \boldsymbol{\theta}_{s'})p(\hat{\mathbf{y}}_{t'}|\mathbf{x}_{t'}, \boldsymbol{\theta}_{s'}, \mathbf{X}_{t'})d\hat{\mathbf{y}}_{t'}d\boldsymbol{\theta}_{t'}. \tag{16}$$

Theoretically, the distribution $p(\boldsymbol{\theta}_{t'})$ is obtained by $p(\boldsymbol{\theta}_{t'}|\mathbf{y}_{t'}, \mathbf{x}_{t'}, \boldsymbol{\theta}_{s'}) \propto p(\mathbf{y}_t|\mathbf{x}_t, \boldsymbol{\theta}_t)p(\boldsymbol{\theta}_t|\boldsymbol{\theta}_s)$, where $p(\boldsymbol{\theta}_t|\boldsymbol{\theta}_s)$ is the prior distribution. To simplify the formulation, we approximate the integration of $p(\boldsymbol{\theta}_{t'})$ by the maximum a posterior (MAP) value of $\boldsymbol{\theta}_{t'}^*$. We obtain the MAP value by training the model $\boldsymbol{\theta}$ with inputs $\mathbf{x}_{t'}$ and pseudo labels $\mathbf{y}_{t'}$ starting from $\boldsymbol{\theta}_{s'}$. The formulation then is derived as:

$$p(\mathbf{y}_{t'}|\mathbf{x}_{t'}, \boldsymbol{\theta}_{s'}, \mathbf{X}_{t'}) = \int p(\mathbf{y}_{t'}|\mathbf{x}_{t'}, \boldsymbol{\theta}_{t'}^*) \int p(\hat{\mathbf{y}}_{t'}|\mathbf{x}_{t'}, \boldsymbol{\theta}_{s'}, \mathbf{X}_{t'})d\hat{\mathbf{y}}_{t'}. \tag{17}$$

To obtain better pseudo labels $\hat{\mathbf{y}}_{t'}$, we further introduce the latent variable $\mathbf{w}_{t'}$ into eq. (17) and a variational posterior of the joint distribution $q(\hat{\mathbf{y}}_{t'}, \mathbf{w}_{t'})$. The formulation is then derived as:

$$
\begin{aligned}
&p(\mathbf{y}_{t'}|\mathbf{x}_{t'}, \boldsymbol{\theta}_{s'}, \mathbf{X}_{t'}) \\
&= \int \int p(\mathbf{y}_{t'}|\mathbf{x}_{t'}, \boldsymbol{\theta}_{t'}^*) \int p(\hat{\mathbf{y}}_{t'}, \mathbf{w}_{t'}|\mathbf{x}_{t'}, \boldsymbol{\theta}_{s'}, \mathbf{X}_{t'})d\hat{\mathbf{y}}_{t'}d\mathbf{w}_{t'} \\
&= \int \int p(\mathbf{y}_{t'}|\mathbf{x}_{t'}, \boldsymbol{\theta}_{t'}^*) \int \frac{q(\hat{\mathbf{y}}_{t'}, \mathbf{w}_{t'}|\mathbf{x}_{t'}, \boldsymbol{\theta}_{s'}, \mathbf{X}_{t'}, \mathbf{Y}_{t'})}{p(\hat{\mathbf{y}}_{t'}, \mathbf{w}_{t'}|\mathbf{x}_{t'}, \boldsymbol{\theta}_{s'}, \mathbf{X}_{t'})}p(\hat{\mathbf{y}}_{t'}, \mathbf{w}_{t'}|\mathbf{x}_{t'}, \boldsymbol{\theta}_{s'}, \mathbf{X}_{t'})d\hat{\mathbf{y}}_{t'}d\mathbf{w}_{t'},
\end{aligned} \tag{18}
$$

where $q(\hat{\mathbf{y}}_{t'}, \mathbf{w}_{t'}|\mathbf{x}_{t'}, \boldsymbol{\theta}_{s'}, \mathbf{X}_{t'}, \mathbf{Y}_{t'})$ and $p(\hat{\mathbf{y}}_{t'}, \mathbf{w}_{t'}|\mathbf{x}_{t'}, \boldsymbol{\theta}_{s'}, \mathbf{X}_{t'})$ denote the prior and posterior distributions, respectively.

## B   IMPLEMENTATION DETAILS

Our train setup follows Iwasawa & Matsuo (2021). We use a batch size of 70 and train our method using the ERM algorithm Gulrajani & Lopez-Paz (2020). As stated, our backbones such as ResNet-18 and ResNet-34 are pretrained on ImageNet same as the previous methods. During our training, the model with highest validation accuracy is selected for adaptation on the target domain. We use similar settings for all Domain Generalization benchmarks that have been reported in the paper. We train all our models on NVIDIA Tesla 1080Ti GPU. In Table 8 we report the runtime required for test-time adaptation. We observe that our method is comparable to methods that are not optimization free such as SHOT and PL. We also in Table 7 report the amount of time consumed during source training.

## C   DETAILED EXPERIMENTAL RESULTS

**Detailed experimental results.**   In Table 4 we report our detailed performance and comparison to existing methods on PACS with both ResNet-18 and ResNet-50 as the backbone. We observe that our method shows an improvement in accuracy compared to other methods especially on "Art-Painting", "Cartoon" and "Photo" Domains. The results show the benefits of using Variational Pseudo labels for adaptation.

We also conduct experiments on rotated MNIST and rotated Fashion-MNIST for comparison as shown in Table 5. We follow the settings in Piratla et al. (2020) and use ResNet-18 as the backbone. The conclusion is similar to that in PACS. Our method achieves better performance than both the non-adaptive domain generalization methods and adaptation methods.

Table 4: **Comparisons on PACS.** The experiments are conducted on PACS averaged over five runs. We provide the results of our method under both the online and offline adaptation settings. Our method is better than other methods on both settings with different backbones. Our method also performs better than the state-of-the-art domain generalization methods.

| Algorithm | Photo | Art-painting | Cartoon | Sketch | *Mean* |
|---|---|---|---|---|---|
| *ResNet-18* | | | | | |
| *Online adaptation* | | | | | |
| Tent-BN (Wang et al., 2021) | 93.9 ±0.3 | 78.5 ±0.8 | 78.3 ±0.4 | 74.4 ±0.2 | 81.3 ±0.3 |
| T3A (Iwasawa & Matsuo, 2021) | 94.7 ±0.5 | 80.4 ±0.7 | 75.2 ±0.4 | **76.5** ±0.2 | 81.7 ±0.4 |
| TAST (Jang & Chung, 2022) | **96.4** ±0.2 | 80.6 ±0.5 | 78.3 ±1.0 | 72.5 ±0.8 | 81.9 ±0.4 |
| SHOT (Liang et al., 2020) | 96.2 ±0.3 | 81.1 ±0.9 | 79.7 ±0.9 | 72.5 ±2.0 | 82.4 ±0.6 |
| *Our Method* | 95.5 ±0.2 | **82.9** ±0.4 | **81.3** ±0.5 | 74.1 ±0.7 | **83.4** ±0.2 |
| *Offline adaptation* | | | | | |
| Tent-BN (Wang et al., 2021) | 94.6 ±0.2 | 80.1 ±0.9 | 79.4 ±0.8 | **75.0** ±1.0 | 82.3 ±0.3 |
| *Our Method* | **95.8** ±0.5 | **84.3** ±0.8 | **83.4** ±0.4 | 74.6 ±0.2 | **84.8** ±0.4 |
| *ResNet-50* | | | | | |
| *Online adaptation* | | | | | |
| Tent-BN (Wang et al., 2021) | 96.0 ±0.4 | 84.9 ±0.4 | 79.8 ±0.6 | 75.7 ±0.7 | 84.1 ±0.3 |
| T3A (Iwasawa & Matsuo, 2021) | 96.4 ±0.2 | 86.0 ±0.6 | 80.3 ±0.9 | 75.2 ±1.5 | 84.5 ±0.3 |
| SHOT (Liang et al., 2020) | 96.5 ±0.5 | 84.6 ±1.7 | 80.1 ±1.3 | 74.8 ±2.9 | 84.1 ±1.0 |
| TAST (Jang & Chung, 2022) | 96.9 ±0.5 | 83.8 ±0.5 | 79.1 ±0.5 | **76.4** ±0.5 | 84.1 ±0.5 |
| *Our Method* | **96.9** ±0.2 | **87.0** ±0.4 | **83.8** ±0.3 | 74.3 ±0.2 | **85.5** ±0.3 |
| *Offline adaptation* | | | | | |
| Tent-BN (Wang et al., 2021) | 96.5 ±0.2 | 85.8 ±0.3 | 81.5 ±0.7 | **76.9** ±0.7 | 85.1 ±0.3 |
| *Our Method* | **97.6** ±0.2 | **87.3** ±0.4 | **84.2** ±0.2 | 75.3 ±0.3 | **86.1** ±0.2 |

Table 5: **Comparison on rotated MNIST and Fashion-MNIST.** The models are evaluated on the test sets of MNIST and Fashion-MNIST with rotation angles of $0°$ and $90°$. Our method performs better than both non-adaptive domain generalization methods (Dou et al., 2019; Piratla et al., 2020) and adaptive methods (Wang et al., 2021; Xiao et al., 2022).

| | MNIST | Fashion-MNIST |
|---|---|---|
| Dou et al. (2019) | 93.2 | 72.4 |
| Piratla et al. (2020) | 94.7 | 78.0 |
| Wang et al. (2021) | 95.3 | 78.9 |
| Xiao et al. (2022) | **95.8** | 80.8 |
| *Our method* | **95.9** ±0.1 | **82.4** ±0.2 |

**Extra ablations**  We also provide extra ablation studies on the pseudo label generation and adaptation with our variational pseudo labels. We directly make predictions using the variational pseudo labels that are generated by sampling from the pseudo label distributions at test time. As shown in Table 6, the predictions based on the pseudo-label distributions are better than the ERM baseline, demonstrating that our variational pseudo labels are better than the original pseudo labels. Moreover, after adapting the model parameters by our variational pseudo labels, the performance further improves obviously.

Table 6: **Investigating by evaluating directly using the pseudo label distributions.** We conduct the below experiments on PACS using ResNet-18. Our method that adapts the models with variational pseudo labels performs better than the prediction by directly sampling from the pseudo label distributions.

| Settings | **Photo** | **Art-painting** | **Cartoon** | **Sketch** | *Mean* |
|---|---|---|---|---|---|
| Prediction by pseudo label distributions | 95.97 ±1.0 | 81.23 ±0.5 | 79.19 ±0.7 | 73.22 ±0.5 | 82.40 ±0.55 |
| *Our method* | **95.80** ±0.5 | **84.32** ±0.8 | **83.44** ±0.4 | 74.57 ±0.2 | **84.78** ±0.36 |

Table 7: **Runtime required for source training on PACS using ResNet-18 as a backbone network.** The proposed method has larger time costs during training due to the meta-learning strategy but introduces few extra parameters.

|  | Parameters | Time for 10000 iterations |
|---|---|---|
| ERM | 11.18M | 6.5 hours |
| *Our method* | 11.96M | 14.6 hours |

Table 8: **Runtime averaged for datasets using ResNet-18 as a backbone network.** The proposed method has similar or even better time costs at test time with the other test-time adaptation methods.

|  | VLCS | PACS | Terra-Incognita |
|---|---|---|---|
| Tent (Wang et al., 2021) | 7m 28s | 3m 16s | 10m 34s |
| Tent-BN (Wang et al., 2021) | 2m 8s | 33s | 2m 58s |
| SHOT (Liang et al., 2020) | 8m 9s | 4m 22s | 12m 40s |
| T3A (Iwasawa & Matsuo, 2021) | 2m 9s | 33s | 2m 59s |
| TAST (Jang & Chung, 2022) | 10m 34s | 9m 30s | 26m 14s |
| *Our method* | 2m 20s | 5m 33s | 14m 30s |

**Time cost analyses.** We also provide the time cost of our method in both the training (Table 7) and inference stage (Table 8). As we utilize the meta-learning strategy to learn the ability to handle domain shifts during training, the time cost during training is larger than the ERM baseline. Moreover, compared with the ERM baseline, our variational pseudo-label learning and meta-learning framework only introduce a few parameters. Since the meta-learning strategy only complicates the training process, our method has similar runtime in inference compared with the other test-time adaptation methods, e.g., Tent (Wang et al., 2021) and source-free domain adaptation methods, e.g., SHOT (Liang et al., 2020).

