# OpenReview forum: "Variational Pseudo Labels for Meta Test-time Adaptation"
_ICLR.cc/2023/Conference — Submitted to ICLR 2023_

### Official Review · Reviewer_6yHt · 2022-10-25

**Confidence:** 4
**Correctness:** 3
**Technical Novelty And Significance:** 2
**Empirical Novelty And Significance:** 2
**Recommendation:** 5

**Clarity, Quality, Novelty And Reproducibility:**

The overall idea is relatively clearly presented. But some key ingredients in meta-learning are missing which makes it hard judge whether these operations are fair. For example, how is the domain shift simulated on source domain data is a very important concern. The idea of sampling pseudo labels from This method can be hardly reproduced if source code is not released.


**Strength And Weaknesses:**

Strength:

1. Allow pseudo label sampling is a novel idea in TTA.

Weakness:

1. Experiments are not carried out on commonly adopted TTA datasets, e.g. CIFAR10-C, CIFAR100-C, ImageNet-C, VisDA. I would like to know why benchmarking on the above datasets are missing.

2. Meta learning requires a held-out validation set. Does it require the validation set to mimic the potential corruptions or distribution shift in the real target domain? If such knowledge is available does it mean the domain shift is a known priori? I am afraid this violates the assumptions of TTA where the information on target domain should be kept unknown before inference.

3. Since the benefit is potentially contributed by sampling pseudo label from a distribution. It is recommended to evaluate directly sampling pseudo label from the posterior, i.e. the probablistic output of unlabeled data, on the presented datasets.

4. I am wondering how integrating the features of neighboring target samples can be achieved in an online TTA mode. If test samples arrive in a stream neighboring samples can not be collected before they are seen.

5. Is the meta-learning adaptation a necessary step? If not, it is recommended to separately evaluate the importance of incorporating the meta-learning adaptation.

6. One important concern for test-time training is the inference speed. How much extra computation overhead does this variational pseudo label introduce?




**Summary Of The Paper:**

This paper addressed test-time adaptation by incorporating uncertainty. This is inspired by the fact that pseudo labels could be confidently wrong, thus pseudo labels are treated as a distribution and test-time training is carried out on the pseudo labels sampled from the distribution. A meta-learning approach is further proposed to better handle domain shifts between source and target domains.


**Summary Of The Review:**

Overall, this paper presents some interesting ideas for test-time adaptation. In particular, sampling pseudo labels from categorical distribution is interesting, but awaits more empirical evidence. The meta-learning approach is not clearly explained thus potentially raising the question of whether domain shift must be a prior knowledge. Some commonly adopted experiment settings are missing in this work which limits evaluating the proposed method.

---

> ### Author Response · Authors · 2022-11-18
> **Response to Reviewer 6yHt**
>
> **Weakness:**
>
> **1. Experiments on DA datasets**
>
> Our technique relies on multi-source domain setting to learn domain shifts during training process. With datasets quoted, inherently the domain shift doesnt exist. We will mitigate this problem.
>
> **2. Clarification about experiment settings**
>
> We would like to clarify that our method does not violate the assumption of test-time adaptation. During training time, we split the source domains into meta-source and meta-target domains, which helps us use a held-out validation set. We do not use any target data during training since our model is already exposed to domain shifts in the training stage.  At test time we directly sample variational labels and use it.
>
> **3. Evaluating directly using probabilistic labels on test data**
>
> We have also added the results for this setting in section B of appendix.
>
> **4.Neighbors in online mode**
>
> During test time, a source-trained model that has already been exposed to domain shifts is being used. At Test-time, similar to existing methods the data influx is in the form of batches, hence the idea of integrating features of neighbor target samples can still be leveraged.
>
> **5. Importance of meta learning**
>
> Our method proposes  to learn the process of generating variational pseudo labels. We would like to clarify that during source training meta-learning helps to learn the ability to learn and correct the pseudo labels by variational inference. Specifically, since we have access to true labels in the meta-source and meta-target stage,  we minimize the KL Divergence between the predicted label and true label. At test time, since our model is already exposed to domain shifts we directly sample the pseudo labels from the prior and use them for training.  Following your suggestion, we have also added the results for variational labels without meta-learning in main paper Section 4.2.
>
>
> **6. Inference speed**
>
> We have added the runtime consumed during test time adaptation and also the time consumed during source training in the appendix Section B.
>
> | Method     | VLCS    | PACS   | TerraIncognita |
> |------------|---------|--------|----------------|
> | Tent       | 7m 28s  | 3m 16s | 10m 34s        |
> | TentBN     | 2m 8s   | 33s    | 2m 58s         |
> | SHOT       | 8m 9s   | 4m 19s | 12m 40s        |
> | PL         | 8m 18s  | 4m 43s | 12m 55s        |
> | T3A        | 2m 9s   | 33s    | 2m 59s         |
> | TAST       | 10m 34s | 9m 30s | 26m 14s        |
> | Our Method | 2m 20s  | 5m 33s | 14m 30s        |

---

### Official Review · Reviewer_vang · 2022-10-26

**Confidence:** 3
**Correctness:** 3
**Technical Novelty And Significance:** 2
**Empirical Novelty And Significance:** 2
**Recommendation:** 3

**Clarity, Quality, Novelty And Reproducibility:**

The formulation is original. However, I have several concerns with this formulation.

This also hinders the clarity of the paper. I suggest the authors to address these concerns about the probabilistic formulation to improve both the correctness and clarity of the paper.

**Strength And Weaknesses:**

**Strength**: The paper discusses an interesting view of the online adaptation process (the variational meta-learning setting).

**Weakness**:
- I have several concerns about the probabilistic view of the adaptation process used in the paper.
- The empirical evaluation is not weak.

Details of these weaknesses can be found below.

**Summary Of The Paper:**

The paper discusses a probabilistic view of the adaptation process. Based on that, it introduces a meta-adaptation setting with a variational pseudo-labeling distribution.

**Summary Of The Review:**

**Concerns about the formulation**
- I have some concerns about the probabilistic formulation of the paper. For example, the authors suggest that the distributions $p(\theta_t|x_t,\theta_s)$ or $p(\theta_t|\hat{y}_tx_t,\theta_s)$ are posterior distributions. Can the authors clarify and elaborate? What is the prior distribution (of $\theta_t$)? We can't really have a posterior distribution without talking about the prior.
- I suggest the authors discuss in detail the generating process (all the priors and likelihood). Meaning, to discuss a joint distribution over $\theta_t,\theta_s,x_t,x_s,y_s,y_t$ as well as any other latent variables such as $w_t$, and all the terms that this joint distribution factorized into. Then, condition on the observed variables, we can discuss the posterior distributions and predictive distribution. This would make the paper much clearer and help us understand the formulation more deeply.

**Concerns about the evaluation**
The evaluation in the paper seems incomprehensive.
- First of all, it is beneficial to validate the method on standard online domain adaptation datasets, such as Cifar10/Cifar100/ImageNet Corruption and the VisDA17 dataset. This will help us to have an idea of how the method compares against the literature.
- I also suggest the authors include stronger baselines. For example, EATA [1], TTT[2] or TTT++[3]


[1]  Niu, S., Wu, J., Zhang, Y., Chen, Y., Zheng, S., Zhao, P. &amp; Tan, M.. (2022). Efficient Test-Time Model Adaptation without Forgetting. Proceedings of the 39th International Conference on Machine Learning
[2] Yu Sun, Xiaolong Wang, Zhuang Liu, John Miller, Alexei Efros, and Moritz Hardt. Test-time training with self-supervision for generalization under distribution shifts. In International conference on machine learning, pages 9229–9248. PMLR, 2020.
[3] Yuejiang Liu, Parth Kothari, Bastien van Delft, Baptiste Bellot-Gurlet, Taylor Mordan, and Alexandre Alahi.TTT++: When does self-supervised test-time training fail or thrive? Advances in Neural Information Processing Systems, 2021.

---

> ### Author Response · Authors · 2022-11-18
> **Response to Reviewer vang**
>
> **Formulation**
>
> We regret not clarifying the formulation. We now provide it here.
> The distribution $p(\theta_t|\mathbf{y}_t,  \mathbf{x}_t, \theta_s) \propto p( \mathbf{y}_t |  \mathbf{x}_t, \theta_t) p(\theta_t| \theta_s)$. Therefore, we treat $p(\theta_t| \theta_s)$ as the prior distribution of $\theta_t$.
>
> We start the objective from the conditional distribution  $p(\mathbf{y}_t | \mathbf{x}_t, \theta_s, \mathbf{X}_t)$.
> Here we provide the detailed generating process of the formulation:
>
> $$ p(\mathbf{y}_t| \mathbf{x}_t, \theta_s, \mathbf{X}_t)  = \int p(\mathbf{y}_t|\mathbf{x}_t, \theta_t) p(\theta_t |\mathbf{x}_t, \theta_s, \mathbf{X}_t) d \theta_t. $$
>
> We then introduce the pseudo labels $\hat{\mathbf{y}}_t$ as the latent variable into the above equation and derive it as:
>
> $$ p(\mathbf{y}_t | \mathbf{x}_t, \theta_s, \mathbf{X}_t) = \int p(\mathbf{y}_t|\mathbf{x}_t, \theta_t) \int p(\theta_t|\hat{\mathbf{y}}_t, \mathbf{x}_t, \theta_s) p(\hat{\mathbf{y}}_t | \mathbf{x}_t, \theta_s, \mathbf{X}_t) d \hat{\mathbf{y}}_t d \theta_t. $$
>
> Theoretically, the distribution $p(\theta_t)$ is obtained by $p(\theta_t|\mathbf{y}_t, \mathbf{x}_t, \theta_s) \propto p( \mathbf{y}_t |  \mathbf{x}_t, \theta_t) p(\theta_t| \theta_s)$, where $p(\theta_t| \theta_s)$ is the prior distribution.
> To simplify the formulation, we approximate the integration of $p(\theta_t)$ by the maximum a posterior (MAP) value $\theta_t^*$.
> We obtain the MAP value by training the model $\theta$ with inputs $\mathbf{x}_t$ and pseudo labels $\mathbf{y}_t$ starting from $\theta_s$. The formulation then is derived as:
>
> $$  p(\mathbf{y}_t| \mathbf{x}_t, \theta_s, \mathbf{X}_t) = \int p(\mathbf{y}_t|\mathbf{x}_t, \theta_t^*) \int p(\hat{\mathbf{y}}_t | \mathbf{x}_t, \theta_s, \mathbf{X}_t) d \hat{\mathbf{y}}_t. $$
>
> To obtain better pseudo labels $\hat{\mathbf{y}}_t$, we further introduce the latent variable $\mathbf{w}_t$ into the formulation and design a variational posterior of the joint distribution $q(\hat{\mathbf{y}}_t, \mathbf{w}_t)$. The formulation is then derived as:
>
> $$ p(\mathbf{y}_t| \mathbf{x}_t, \theta_s, \mathbf{X}_t) = \int \int p(\mathbf{y}_t|\mathbf{x}_t, \theta_t'^*) \int p(\hat{\mathbf{y}}_t, \mathbf{w}_t' | \mathbf{x}_t, \theta_s, \mathbf{X}_t) d \hat{\mathbf{y}}_t d \mathbf{w}_t  = \int \int p(\mathbf{y}_t|\mathbf{y}_t, \theta_t^*) \int \frac{q(\hat{\mathbf{y}}_t, \mathbf{w}_t | \mathbf{x}_t, \theta_s, \mathbf{X}_t, \mathbf{Y}_t)}{p(\hat{\mathbf{y}}_t, \mathbf{w}_t | \mathbf{x}_t, \theta_s, \mathbf{X}_t)} p(\hat{\mathbf{y}}_t, \mathbf{w}_t | \mathbf{x}_t, \theta_s, \mathbf{X}_t) d \hat{\mathbf{y}}_t d \mathbf{w}_t, $$
>
> where $q(\hat{\mathbf{y}}_t, \mathbf{w}_t | \mathbf{x}_t, \theta_s, \mathbf{X}_t, \mathbf{Y}_t)$ and $p(\hat{\mathbf{y}}_t, \mathbf{w}_t | \mathbf{x}_t, \theta_s, \mathbf{X}_t)$ denote the prior and posterior distributions, respectively.
> We also added these to the Appendix.
>
> **Evalutation**
>
> Since we demonstrate our method in an online setting we choose baselines that have been demonstrated in the aforementioned setting. In terms of comparison, we have also added the baselines that are quite recent and also include an unpublished paper. The baselines that we compare to our method are recent and more relevant to our method’s formulation. Our technique relies on the multi-source domain setting to learn domain shifts during the training process. With datasets quoted, inherently the domain shift does not exist. We will mitigate this problem.

---

### Official Review · Reviewer_ZtPj · 2022-10-27

**Confidence:** 4
**Correctness:** 2
**Technical Novelty And Significance:** 3
**Empirical Novelty And Significance:** 2
**Recommendation:** 3

**Clarity, Quality, Novelty And Reproducibility:**

[Clarity and Quality]
This paper is generally well written. However, there are some confusing parts:
- What is the test-time adaptation in table 1?
- Contents of Table 1 and Table 2 seems to be largely overwrapped.

[Originality]
The analysis and derived meta-learning based algorithms to compute variational pseudo label is novel.

[Reproducibility]
Since the paper is mainly based on empirical results, it is preferable to provide full access to experimental code.


**Strength And Weaknesses:**

[Strength]
(1) Test-time adaptation is timely and relevant topic for the conference.
(2) The paper is well written and easy to follow.
(3) Meta-learning based algorithms with pseudo label is novel and interesting.


[Weaknesses]
(1) Empirical validation is limited to small-scale dataset. While I agree PACS and rotated MNIST are standard benchmarks, but there are lot of more benchmark dataset (e.g., DomainNet, OfficeHome, TerraIncognita, etc. [1]). Also, it lacks comparison with standard DG methods. Since the standardized evaluation is regarded as one of the big issue in DG papers [1] and this paper does not provide theoretical results, I strongly recommend adding more empirical evaluations to fully show the benefit of the proposal.

[1] Gulrajani, Ishaan and David Lopez-Paz. “In Search of Lost Domain Generalization.” ICLR2021

(2) There are some missing literatures about pseudo labeling both inside [2, 3] and outside [4] the field of the test-time adaptation. Since the pseudo label is the core of the paper, authors should discuss the differences between these studies. Besides, it is also preferable to add these methods as baselines.

[2] Goyal, Sachin et al. “Test-Time Adaptation via Conjugate Pseudo-labels.” NeurIPS2022
[3] Wang, Qin, Olga Fink, Luc Van Gool, and Dengxin Dai. 2022. “Continual Test-Time Domain Adaptation.” CVPR2022
[4] Rizve, Mamshad Nayeem et al. “In Defense of Pseudo-Labeling: An Uncertainty-Aware Pseudo-label Selection Framework for Semi-Supervised Learning.” ICLR2021

**Summary Of The Paper:**

This paper presents a test-time adaptation (TTA) method from probabilistic perspective. Specifically, they analyze first analyze the pseudo labeling, which is a naive approach in TTA, from probabilistic point of view. Based on the analysis, they propose variational pseudo labels,　and meta-learning based algorithms. The empirical validation is conducted on PACS, rotated MNIST and Fashion-MNIST.


**Summary Of The Review:**

While I agree that the pseudo-label in TTA is worth investigating and derived meta-learning based algorithms is interesting, the paper in current form lacks comprehensive empirical results and discussion about the relationships with prior works.

---

> ### Author Response · Authors · 2022-11-18
> **Response to Reviewer ZtPj**
>
> **Weaknesses**
>
> **1. Empirical evaluation on other datasets**
>
> We provide the results on two more widely-used datasets such as VLCS and TerraIncognita in domain generalization and more comparisons with the standard domain generalization methods.
>
> Our method performs better than both domain generalization and test-time adaptation methods.
> We added these results in the paper.
>
>  |           | PACS             |           | VLCS      | TerraIncognita |
> |------------------|-----------|-----------|-----------|----------------|
> |                  | ResNet-18 | ResNet-50 | ResNet-18      | ResNet-18 |
> | Dou et al. 2019  | 81.04     | 82.67     | -              | -         |
> | Zhao et al. 2020 | 81.46     | 85.34     | 74.38          | -         |
> | ERM              | 80.3      | 85.7      | 73.2           | 40.7      |
> | Tent             | 80        | 79.6      | 71.5           | 40        |
> | T3A              | 81.7      | 84.5      | 76.5           | 41.6      |
> | Ours             | 83.4      | 85.5      | 77.8           | 46.2      |
>
> **2. Missing literature review**
>
> We have added the papers suggested to our related works and discussed the differences between them. To summarize, [2] aims to meta-learn the loss function and relies on the labels produced by the corresponding loss, [3] utilizes confidence to approximate the domain difference and applies augmentations to improve quality if below the threshold to update the student model. However, both do not rely on addressing uncertainty using variational methods to approximate them as distributions and also the ability to learn such label generation using meta-learning. [4] Assumes that there is access to target labels in a semi-supervised learning and relies on prediction uncertainties leveraged through the labeled samples in target to and generates new pseudo labels.
>
> **Clarity, Quality, Novelty And Reproducibility:**
>
> **1. What is test time adaptation in table 1?**
>
> The test-time adaptation in table 1 is Tent (Wang et al. 2020). We clarified this in the paper.
>
> **2. Contents of table 1 and table 2**
>
> We updated Table 2 in the main paper and added more comparisons on more datasets with more methods. The detailed comparisons on PACS is moved to Appendix.
> Reproducibility - We will provide the full source code.

---

### Official Review · Reviewer_vwGZ · 2022-10-27

**Confidence:** 4
**Correctness:** 3
**Technical Novelty And Significance:** 2
**Empirical Novelty And Significance:** 3
**Recommendation:** 5

**Clarity, Quality, Novelty And Reproducibility:**

*Clarity*:

The exposition is clear, the writing straightforward, and the organization of the work is easy to navigate and digest.
The setting and method are clearly explained by the notation of Sections 2.1 and 2.2 and Figure 1.
The intuition for why sampling and variational pseudo-labels should help is communicated with a toy example.
However, the implementation details leave some points to be desired, especially given that test-time adaptation methods can be sensitive to hyperpareters like the batch size and number of updates.


*Quality*:

- The experiments and exposition are well-executed under the chosen scope, but that scope is limited. There are more possible benchmarks, and these benchmarks are common in the prior papers on this subject.
- The overall accuracy improvements are small, and in line with improvements from better tuning of learning rates and batch sizes for methods like Tent. See https://arxiv.org/abs/2104.12928 and results for "ENT (ours)".
  The improvement due to the meta-learning of variational pseudo-labels, which is the main technical contribution of the paper, is smaller still (see Table 1).

*Novelty*:

- Probabilistic sampling of pseudo-labels is straightforward but nevertheless novel. However, it is not quite correct to say that this is the first way to leverage uncertainty. The updates of soft pseudo-labeling and entropy minimization in the style of Tent both change as a function of the confidence of a pseudo-label.
- Variational pseudo-labeling as defined and implemented here is novel. Other methods like SHOT (by prototypes) and TAST (by neighbors and prototypes) also make use of other points in defining their pseudo-labels, but do not meta-learn a pseudo-labeling inference network as done by the proposed work.
- Variational pseudo-labeling approach is not the first to make use of other points in the batch. In fact, most test-time methods do, because their batch-wise statistics or parameter updates couple inference across inputs.

*Reproducibility*:

There are some gaps in the implementation details that would take some effort and experimentation to determine. There is no statement about providing the code.



**Strength And Weaknesses:**

*Strengths*

- The topic of pseudo-label generation is relevant and applicable to the improvement of test-time adaptation, as a set of existing methods adapt by entropy minimization and/or pseudo-label optimization.
- The meta-learning approach makes sense, and while there is generality in not requiring specialized training (as other methods like BN and Tent do not), if better accuracy can be achieved with different training then it is worthwhile to pursue it.
- The exposition of the main idea and its technical details is clear (but see weaknesses for missing implementation detail).
- The related work provides and apt and concise summarization of published test-time adaptation methods.
- The evaluation experiments with the standard ResNet-50 and the smaller ResNet-18 to show that results generalize beyond a single architecture (although these two are related).
- It is nice to see that the proposed variational pseudo-labels improve the accuracy at most steps of adaptation (Figure 2, right) and not only at the end of optimization (Figure 2, left).

*Weaknesses*

- Related work is missing, specifically confidence regularized self-training https://arxiv.org/abs/1908.09822 that likewise addresses the uncertainty and error in pseudo-labeling.
- The method is not fully described in its implementation: what are the batch sizes, number of updates, and the choice of parameters?
  (The parameters likely follow those of Tent and other methods, which update the affine parameters in normalization layers, but this should be specified.)
- The amount of improvement is limited. On the main evaluation dataset of PACS, the improvement is about +2 points for online or offline adaptation against Tent (a common baseline).
- The evaluation is narrower than most work on test-time adaptation. In particular, the works compared against include more benchmarks, such as ImageNet-C for image corruption (Tent, TAST, SHOT), and VLCS + OfficeHome + Terra Incognita for domain generalization (T3A). Relative to these established options, the chosen Fashion-MNIST and MNIST are toy datasets.
- The process for model and hyperparameter selection is not explained. How were the learning rates for optimization chosen, for example? Are all of these tuned to the accuracy of adaptation on the test set?
- The training could be more computationally intensive per step, and may require more steps overall, due to the multiple phases of meta-learning. This is not discussed or quantified.
  The meta-learning approach is likely more expensive than standard supervised training, but the question is how much? This should be measured empirically.



**Summary Of The Paper:**

Test-time adaptation updates a model during testing to improve its generalization to different data.
A popular choice of test-time optimization is entropy minimization or pseudo-labeling, which trains on the model's own predictions.
This work seeks to prepare a model for such test-time optimization by meta-learning (1) the model parameters for a better initialization and (2) the pseudo-labeling inference to provide better targets for the loss.
As a first step, a probabilistic alternative to deterministic pseudo-labels is proposed, and then a variational distribution is defined to condition on the test inputs and their relationships.
The probabilistic pseudo-labels simply sample new pseudo-labels given the pseudo-label probabilities.
The variational distribution for pseudo-labels conditions on the given input and the other samples in the batch through a latent weighting $w$.
Training has three phases: meta-source optimization learns the source model parameters, meta-adaptation infers the variational pseudo-labels and optimizes the target model parameters, and then meta-target optimization updates the source model parameters by the meta-loss to achieve better meta-adaptation optimization.
At the same time, the variational pseudo-labeling parameters are jointly updated alongside the model parameters, according to the meta-loss and a cross-entropy loss between the pseudo-labels and true labels.
Testing proceeds as usual for test-time adaptation, with alternating predictions and updates, as shown by Eqs. 10 and 14.
That is, given the meta-learned model and pseudo-labeling inference, the model parameters are updated during testing according to the cross-entropy loss on the predicted pseudo-labels.
Experiments evaluate adaptation on PACS, a common domain generalization dataset, along with toy datasets based on MNIST with artificial domains made by rotating images, while comparing with recent test-time adaptation methods.
Accuracy improves by 1-2 points with the proposed probabilistic pseudo-labels and variational pseudo-labels (Table 1, Table 3, Table 4).
Analysis experiments show that the variational pseudo-labels reliably improve a little bit on standard pseudo-labels: overall (Figure 2, left), across iterations (Figure 2, right), and across different amounts of data for adaptation (Figure 3).

**Summary Of The Review:**

This work brings a probabilistic perspective to pseudo-labeling and contributes a meta-learning scheme for training the model parameters and a pseudo-label predictor.
The proposed stochastic and variational pseudo-labels do seem to improve marginally on PACS, but the improvement is only marginal, and the evaluation does not cover other standard choices of test-time adaptation benchmark.
The small effect and narrow evaluation dampen the significance of the proposed techniques even if they are novel variations on pseudo-labeling.
As there is already work experimenting with exactly how to define pseudo-labels—SHOT, RPL, TAST, and conjugate pseudo-labels—more is needed to advance beyond what has already been done and inform future methods for test-time adaptation.
I encourage the authors to more fully evaluate the proposed method and explore if its improvements compound with other techniques to achieve more significant accuracy improvements.

**Update after response**: The response and revision provided experiments for domain generalization to cover more datasets and more baselines. Furthermore, an ablation justifies the contribution of meta-learning and not merely sampling the pseudo-labels, and the computational cost has been measured. These results reinforce the empirical significance of the work. However, the significance for test-time adaptation is still an issue, because this work only addresses multi-domain/multi-source training, and so does not evaluation on standard benchmarks for adaptation like the ImageNet variants (-C, -R, -V2, ...) or VisDA-C. As such the recommendation is raised to 5 but not higher.

*Points for Rebuttal*

1. Please clarify the implementation of the test-time adaptation step. How many updates are made per batch? What is the batch size?
2. Please measure the training computation relative to other methods. This could be measured by some subset of time, FLOPs per step, or number of steps, for example.
3. Please comment on the lack of results for image corruptions (ImageNet-C, CIFAR-10/100-C) which are provided by most prior papers on test-time adaptation. These experiments would provide an informative and more thorough comparison of the proposed method with the state-of-the-art.
4. Please relate the novelty and claims of this work to those of confidence regularized self-training (see link under Weaknesses).
5. Please relate the proposed method and degree of improvement to the conjugate pseudo-labels method (Goyal et al. 2022). This is not a prior published method, so it does not count against this submission, but comparing to it would make this submission more comprehensive.

*Miscellaneous Feedback*

- 1. Introduction
 - "these two settings" would be clearer to a broader audience if it identifies domain adaptation (which requires targets during training) and generalization (which does not harness targets during testing).
 - The "pseudo-labels are more accurate" are they? are the pseudo-labels more correct in themselves, or better calibrated, or what? they can be more effective, if the adaptation is more accurate, but that does not mean the pseudo-labels are in themselves more accurate
- 4.3 Comparison
  - What about consistency regularization/DIRT-T and VAT? Such methods sample virtual "neighbors" of each point by penalizing changes to the output on small changes to the input. MEMO likewise updates on augmentations of the input. Do the proposed pseudo-labels compound with these methods and give more improvement?

---

> ### Author Response · Authors · 2022-11-18
> **Response to Reviewer vwGZ**
>
> **Weakness**
>
> **1. Related work**
>
> Thank you for mentioning the work. We have added Zou et. al (2019) in the related work.
>
> **2. Detailed listing of implementation steps**
>
> We update the full model in a meta learning setup since we generate pseudo labels that can be used to train the model. We set the batch sizes as 70 and number of updates equivalent to one pass of data. We added all the implementation details in the Appendix.
>
> **3. Limited improvement**
> Even a small improvement is difficult to achieve; for instance, T3A improves Tent by about 1 point. In this sense, we consider our improvement (2 points) to be considerable. In this revised version, we have added more results on more datasets and the results for VLCS and Terra Incgonita. On VLCS, we are better than compared methods by 1.2+ points overall and around 5+ points overall on Terra Incognita dataset.
>
> **4. More datasets**
>
> We have added new results in the updated version of the paper for VLCS and Terra Incognita. We show that we are better than compared methods by a good margin.
>
> **5.  Hyperparameter Section**
>
>  We choose the hyperparameters based on the validation set as mentioned in “In Search of lost domain generalization” and T3A. We have added the specifics in the appendix section.
>
> **6. Computational Costs**
>
> Following your suggestion, we have now added both runtime during test-time adaptation and the training time in comparison to existing related methods. We have also added the tables to the appendix.
>
> **Novelty**
>
> We agree that we are not the first to leverage uncertainty of the pseudo labels. Here we would like to clarify the differences of our method from soft pseudo-labeling or entropy minimization methods. These methods utlize the output logits of softmax as the prediction probability and train the model to directly maximize the logits. By contrast, our method formulates the probability of the pseudo labels as distributions and explicitly leverages the uncertainty by sampling different pseudo labels from the distribution. We added these discussions to the related work.
>
> We would like to also clarify the difference from the batchnorm-based methods. The batch-wise statistics make use of the batch information during feature extraction. By contrast, our variational pseudo labels aim to fully utilize the semantic information of the neighbor samples to generate better pseudo labels. Our method is orthogonal with the batchnorm-based methods. We will clarify these differences in the paper.
>
> **Points for Rebuttal**
>
> **1. Implementation Details**
>
> We have answered above and have added it to the supplementary section.
>
> **2. Computational Cost**
>
> We now report both the adaptation time and the training time required by our method and also added to the appendix section.
>
> For test time adaptation:
> | Method     | VLCS    | PACS   | TerraIncognita |
> |------------|---------|--------|----------------|
> | Tent       | 7m 28s  | 3m 16s | 10m 34s        |
> | TentBN     | 2m 8s   | 33s    | 2m 58s         |
> | SHOT       | 8m 9s   | 4m 19s | 12m 40s        |
> | PL         | 8m 18s  | 4m 43s | 12m 55s        |
> | T3A        | 2m 9s   | 33s    | 2m 59s         |
> | TAST       | 10m 34s | 9m 30s | 26m 14s        |
> | Our Method | 2m 20s  | 5m 33s | 14m 30s        |
>
> Training cost of proposed method:
> | Method     | Parameters | 10000 iterations training time |
> |------------|------------|--------------------------------|
> | ERM        | 11.18M     | 6.5 hours                      |
> | Our Method | 11.96M     | 14.6 hours                     |
>
>
> **3. Additional datasets**
>
> | Method     | VLCS | Terra Incognita |   |
> |------------|------|-----------------|---|
> | Tent-BN    | 61.3 | 39.8            |   |
> | Tent-C     | 71.5 | 40              |   |
> | T3C        | 76.5 | 41.6            |   |
> | Our Method | 77.8 | 46.2
>
> We are better than existing test time domain generalization methods by a realtively good margin. For datasets such as Cifar, our technique relies on multi-source domain setting to learn domain shifts during training process. On datasets like cifar, inherently the domain shift doesnt exist. We will mitigate this problem.
>
> **4. Comparison to confidence regularized self training**
>
> The paper assumes that they have access to target samples xt in the form of unlabelled training and hence approximate it followed by using them to directly to retrain the model. However, due to the presence of the target, they perform this joint learning of optimizing the pseudo labels learned.  We have also discussed this in the related work.
>
> **5. Comparison to one of the works**
>
> As stated above, our method relies on a multi-source domain setting.

---

> > ### Comment · Reviewer_vwGZ · 2022-11-24
> > **Thank you for the thorough response. I have raised my score, but issues remain especially concerning test-time adaptation.**
> >
> > > 1. Related work
> >
> > Thank you for addressing and including the missing related work to resolve this weakness.
> >
> > > 2. Detailed listing of implementation steps
> >
> > Thank you for adding the hyperparameters to the text, for including implementation details in appendix B, and for specifying that the code will be released. This resolves this weakness.
> >
> > > 3. Limited improvement
> > > 4. More datasets
> >
> > Thank you for providing more experiments to evaluate (1) more datasets and (2) more recent baselines. The improvements on T3A and TAST are the most compelling. Although the improvement is still small in some cases, it is larger in others (on Terra Incognita, for example). The point that the range of improvement is similar to that of prior papers, such as T3A, is fair.
> >
> > This partly resolves these weaknesses, but not fully, because the provided evaluations focus on domain generalization but not test-time adaptation. In particular, the popular benchmarks for test-time adaptation are missing, like ImageNet-C/R/V2, and this is because the proposed method requires multiple sources/domains for training.
> >
> > > 5. Hyperparameter Section
> >
> > Following the established hyperparameter setup from T3A and "In Search of lost domain generalization" is valid and resolves this weakness.
> >
> > > 6. Computational Costs
> >
> > Thank you for measuring the computation for training and testing and including these measures in the appendix. To summarize, the training takes ~2x the time and testing takes more time than TENT or T3A but less time than TAST and a comparable amount of time as SHOT. Measuring this and reporting the results—as done in the revision—resolves the weakness for the purposes of this work, because more efficient adaptation is not one of its contributions.
> >
> > **Summary**: The response has addressed the issues with (1) clarity, (2) empirical significance w.r.t. domain generalization, and (3) computation. However, it is still empirically lacking w.r.t. test-time adaptation and common domain adaptation benchmarks, and the proposed method is limited to multi-source/multi-domain training. This limitation was not highlighted well in the submission, and reduces the audience for this work. The response merits a raised score, but is still marginally below acceptance, since it does not actually address test-time adaptation from a single source, as it is commonly framed.

---

> ### Author Response · Authors · 2022-11-18
> **Response to Reviewer vwGZ**
>
>
>
> **Miscellaneous Feedback**
>
> **1. Clarification about settting opted.**
>
> We made the sentence clearer as “However, domain adaptation requires a large number of (unlabeled) target data during training while domain generalization does not consider any target information during generalization at all.”
>
> **2. Clarification aboout pseudo labels being more accurate**
>
> In this context, we state that the variational pseudo labels are better than the regular pseudo labels, which is demonstrated by the experiments in the paper in Figure 2 of main paper.
>
> **3. Comparison**
>
> DIRT aims to construct new source domain sinceit has access to target during source training. VAT aims to make use of unlabelled target for adversarial training. We have discussed both in related work section. We make use of TENT’s approach to design our method. We believe that our method can be compounded with MEMO to assign variational labels to the data augmented samples, which is of future interest.
> ment on PACS dataset**

---

### Decision · Program_Chairs · 2023-01-20

**Decision:**

Reject

**Justification For Why Not Higher Score:**

N/A

**Justification For Why Not Lower Score:**

N/A

**Metareview: Summary, Strengths And Weaknesses:**

The proposed method was deemed enough novel, well justified and clearly presented, with some exceptions.
Main issues relate to the validation phase: limited scope of the experiments, missing comparative analysis (e.g., with domain generalization methods), and the need of validation on more standard datasets (since it is limited to small scale datasets). As for the latter, it was also noted only small increases in performance, not all consistent. Finally, some related work is not quoted, as well as an evaluation of the computational cost.
Some reviewers argued about missing details on some stages of the method asking clarifications, including the probabilistic formulation, and about implementation details for reproducibility.

Authors provide some answers to these concerns, also providing some additional results (a common remarks from all reviewers), but they did not succeed to fully satisfy them, who still remained with ratings below threshold.
For these reasons, this paper cannot be accepted to ICLR 23 in this form.


**Summary Of Ac-Reviewer Meeting:**

N/A